# A Focus on the Role of DSC-PWI Dynamic Radiomics Features in Diagnosis and Outcome Prediction of Ischemic Stroke

**DOI:** 10.3390/jcm11185364

**Published:** 2022-09-13

**Authors:** Yingwei Guo, Yingjian Yang, Fengqiu Cao, Mingming Wang, Yu Luo, Jia Guo, Yang Liu, Xueqiang Zeng, Xiaoqiang Miu, Asim Zaman, Jiaxi Lu, Yan Kang

**Affiliations:** 1College of Medicine and Biological Information Engineering, Northeastern University, Shenyang 110169, China; 2College of Health Science and Environmental Engineering, Shenzhen Technology University, Shenzhen 518118, China; 3Department of Radiology, Shanghai Fourth People’s Hospital Affiliated to Tongji University School of Medicine, Shanghai 200434, China; 4Department of Psychiatry, Columbia University, New York, NY 10027, USA; 5School of Applied Technology, Shenzhen University, Shenzhen 518060, China; 6Engineering Research Centre of Medical Imaging and Intelligent Analysis, Ministry of Education, Shenyang 110169, China

**Keywords:** DSC-PWI, dynamic radiomics features, Lasso, dimension reduction, stroke detection, NIHSS assessment, outcome prediction

## Abstract

Background: The ability to accurately detect ischemic stroke and predict its neurological recovery is of great clinical value. This study intended to evaluate the performance of whole-brain dynamic radiomics features (DRF) for ischemic stroke detection, neurological impairment assessment, and outcome prediction. Methods: The supervised feature selection (Lasso) and unsupervised feature-selection methods (five-feature dimension-reduction algorithms) were used to generate four experimental groups with DRF in different combinations. Ten machine learning models were used to evaluate their performance by ten-fold cross-validation. Results: In experimental group_A, the best AUCs (0.873 for stroke detection, 0.795 for NIHSS assessment, and 0.818 for outcome prediction) were obtained by outstanding DRF selected by Lasso, and the performance of significant DRF was better than the five-feature dimension-reduction algorithms. The selected outstanding dimension-reduction DRF in experimental group_C obtained a better AUC than dimension-reduction DRF in experimental group_A but were inferior to the outstanding DRF in experimental group_A. When combining the outstanding DRF with each dimension-reduction DRF (experimental group_B), the performance can be improved in ischemic stroke detection (best AUC = 0.899) and NIHSS assessment (best AUC = 0.835) but failed in outcome prediction (best AUC = 0.806). The performance can be further improved when combining outstanding DRF with outstanding dimension-reduction DRF (experimental group_D), achieving the highest AUC scores in all three evaluation items (0.925 for stroke detection, 0.853 for NIHSS assessment, and 0.828 for outcome prediction). By the method in this study, comparing the best AUC of F*_t_*_-test_ in experimental group_A and the best_AUC in experimental group_D, the AUC in stroke detection increased by 19.4% (from 0.731 to 0.925), the AUC in NIHSS assessment increased by 20.1% (from 0.652 to 0.853), and the AUC in prognosis prediction increased by 14.9% (from 0.679 to 0.828). This study provided a potential clinical tool for detailed clinical diagnosis and outcome prediction before treatment.

## 1. Introduction

Ischemic stroke is the primary reason for disability and the second-leading cause of death worldwide [1]. The surviving patients are usually accompanied by varying neurological deficits, resulting in impaired living quality and burdened families and society. Good clinical outcomes were proven to be correlated with early vessel recanalization [2,3]. Early warning of stroke and accurate assessment of neurological recovery after treatment will facilitate the early prevention of stroke [4], the selection of individualized treatment plans [5], and the recovery of patients [6], thereby reducing the risk of stroke. Therefore, abnormal brain tissue detection and accurate prognostic status prediction are critical factors in stroke treatment.

Cerebral blood flow (CBF) is an essential physiological parameter to evaluate the state of brain tissue in the clinic. Normal blood flow transmission can provide blood oxygen for brain tissue and maintain a stable brain [7,8]. However, once a cerebral vascular occlusion results in ischemia in a region of brain tissue, the blood flow parameters in that ischemic region differ from those in normal tissue [9,10]. Therefore, CBF parameters have been widely used in the diagnosis and treatment of brain tumors [11,12], stroke [13,14,15], and Alzheimer’s disease [16,17]. Clinically, the clinical images to observe blood flow mainly include ultrasonic doppler (UD), digital subtraction angiography (DSA), computed tomography angiography (CTA), computed tomography perfusion (CTP), perfusion-weighted imaging (PWI), and arterial spin labeling (ASL). Among them, due to the influence of the skull, images from UD are rarely used directly for studying brain diseases. Instead, they are primarily used for blood flow velocity detection in the carotid artery, heart, and other organs. The other images are widely used in stroke diagnosis and the prediction of functional recovery. In perfusion imaging, since the contrast agent is difficult to propagate effectively in the damaged tissue, the signal intensity in this area hardly changes. Therefore, the maximum tissue residual function (Tmax) extracted from PWI or CTA images can be used to detect ischemic stroke lesions (Tmax > 6 s) [18], and the region with a reduction of the relative CBF (rCBF) by 30% compared to that of the symmetric side defined as the core infarct area. In addition, the mismatch between ASL or PWI and diffusion-weighted imaging (DWI) can be used to detect the ischemic penumbra [19,20]. It has been proved that hemodynamic parameters (such as Tmax, CBF, etc.) obtained from medical images have been widely used to detect ischemic stroke lesions. With dozens of consecutively scanned three-dimensional (3D) images, there is more information in perfusion images than in other images. However, due to the difficulty in data processing brought by the vast data amount, the existing algorithms usually used the intermediate parameters calculated from the dynamic susceptibility contrast PWI (DSC-PWI) for clinical analysis rather than directly processing them.

In addition, two factors affecting the rehabilitation of stroke patients are the neurological impairment degree of the patients and the used treatment strategy, and the degree of neurological impairment is one of the influencing factors in deciding the treatment strategy. Therefore, the accurate assessment of neurological impairment in stroke patients is significant for treating and rehabilitating stroke. The main parameter for evaluating the degree of neurological impairment in stroke patients is the National Institutes of Health Stroke Scale (NIHSS) [21]. The NIHSS is obtained through questionnaires and usually includes the following domains: level of consciousness, eye movements, the integrity of visual fields, facial movements, arm and leg muscle strength, sensation, coordination, language, speech, and neglect. Each impairment is scored on an ordinal scale ranging from 0 to 2, 0 to 3, or 0 to 4. Item scores are summed to a total score ranging from 0 to 42 (the higher the score, the more severe the stroke) [22]. Previous studies have explored the association between medical images and NIHSS scores. Generally, stroke patients without vascular occlusion or peripheral occlusion in medical imaging have lower NIHSS and better prognoses. However, Ref. [23] reported that one patient with zero NIHSS might have a stroke. Therefore, NIHSS assessment alone or stroke detection alone may be misdiagnosed. If NIHSS-related information can be obtained based on images and combined with the results of stroke detection, it will be beneficial to assess the severity of patients.

Accurate outcome prediction will assist in customizing personalized treatment plans, reducing the situation of poor recovery, and objectively and accurately evaluating the treatment effect [24]. Several studies have shown that stroke outcomes correlate with clinical text information (CTI) and the parameters computed from medical images. For example, lea-Pereira et al. [25] predicted mortality risk scores during admission for ischemic stroke with CTI, such as age, sex, readmission, and neurological symptoms. Xie et al. [26] used patient information, clinical scores, and volumes of lesion tissue to predict the modified Rankin scale (mRS) in three months. Moreover, Brugnara et al. [27] combined location information for lesions, hypertension, diabetes, dizziness, and physical symptoms to perform the prediction. In addition, Ref. [28] used the neutrophil-lymphocyte ratio to predict the mRS. Although diverse clinical information has consistently been associated with outcomes after ischemic stroke, the usefulness of neuroimaging in predicting outcomes has not been definitively established [29]. In previous studies, the characteristics of the lesion tissue were usually used to predict the outcome of patients, while the overall characteristics of the brain tissue were missing. However, the recovery of neurological function is a reflection of the brain’s overall function, so it is necessary to explore the relationship between the overall characteristics of the brain and prognosis.

The information in medical images is crucial for the prevention, detection, treatment, and outcome prediction of ischemic stroke. Thus, extracting valuable information from medical images is an effective technique in clinical practice. Nowadays, radiomics, an innovative method to quantify high-dimensional features from medical images, is widely used in medical image processing. For example, it is used to investigate tumor heterogeneity [30,31] and in clinical decision support systems to improve treatment decision-making and accelerate advancements of clinical decision support systems in cancer medicine [32,33,34,35,36,37]. However, in the field of stroke, only a few studies have explored the role of radiomics in diagnosing ischemic stroke [38], penumbra-based prognosis assessment [39,40], and functional prediction [41]. Prior studies compared prognostic predictions between different diseases based on lesion characteristics. Few studies have used whole-brain features for clinical analysis. However, the appearance of local lesions will inevitably affect the whole-brain features. Therefore, the role of whole-brain features in diagnosing and treating stroke is of great value.

This study aims to explore the role of the whole-brain dynamic radiomics features (DRF) of DSC-PWI in the diagnosis of ischemic stroke, the assessment of neurological impairments, and outcome prediction. The main contributions lie in the following three aspects.
(1)This study explored the role of DRF in ischemic stroke. First, the radiomics features of 3D images in the time series of DSC-PWI were used to obtain the DRF of the whole brain. Then feature selection and dimensionality reduction methods were used to generate various combinations. Finally, by comparing the effects of multiple features in stroke diagnosis, NIHSS evaluation, and outcome prediction, the clinical value of DRF in stroke treatment and outcome prediction can be proved, providing a potential tool for clinical application.(2)In this study, the DRF of the whole brain were extracted instead of lesion features, which reduced the process of lesion segmentation and saved time for clinical treatment.(3)This study can extract useful features related to the target using feature analysis, reducing the problem of enormous computation costs caused by the direct analysis of a four-dimensional (4D) DSC-PWI image.

## 2. Materials and Methods

Detailed materials and methods are introduced in the following subsections. The materials are described in Section 2.1, and the methods are shown in Section 2.2.

### 2.1. Materials

The Institutional Review Boards approved this retrospective study of Shanghai Fourth People’s Hospital, affiliated with the Tongji University School of Medicine and exempted from informed consent. The datasets in our study were collected by the neurology department of the Shanghai Fourth People’s Hospital, affiliated with the Tongji University School of Medicine, China, from 2013 to 2016. A total of 156 DSC-PWI images from 88 patients were retrospectively reviewed and included. All patients were imaged within 24 h of symptom onset, and 22 patients were screened at least twice during pretreatment and post-treatment. After clinical examination, 78 (50%) DSC-PWI images were diagnosed as ischemic stroke. The primary clinical information includes income NHISS, outcome NHISS, and 90-day mRS. The DSC-PWI image for each patient was scanned on a 1.5T MR scanner (Siemens, Munich, Germany), and Table 1 shows the details.

### 2.2. Methods

The proposed method in this study includes four steps: preprocessing DSC-PWI datasets and computing DRF, feature selection and combination strategy, and evaluating the performance of the four combinations of DRF.

#### 2.2.1. Preprocessing DSC-PWI Datasets and Computing Dynamic Radiomics Features

(A)Registration and smoothing of DSC-PWI datasets
The preprocessing is intended to reduce noise and position deviation impacts. First, it includes registering all of the volumes in the time series, smoothing the voxel in the time series, and splitting the skull and brain tissue. This study corrected the DSC-PWI datasets for potential patient motion by registering all of the volumes in the time series. Then, a triple moving average filter was selected to smooth the data voxel-by-voxel with a 1 × 3 filtering kernel. The registration method was introduced in Refs. [42,43], and the filtering method was used in Ref. [40]. Next, the average 3D image was computed from the first ten 3D images and the last ten 3D images in the registered and smoothed DSC-PWI image. Finally, this study used neuroimaging software package FSL [44] to segment the skull from the average 3D image, and then the mask of brain tissue was obtained for each DSC-PWI image (seen in Figure 1a).

(B)Making ground truth for three evaluation items
This study carried out three evaluation items to evaluate the role of DRF in the diagnosis and outcome prediction of ischemic stroke patients. The three evaluation items were ischemic stroke detection, NIHSS assessment, and outcome prediction. Ischemic stroke detection is used to recognize the presence of ischemic stroke lesions. The NIHSS can reflect the degree of neurological impairment, and the 90-day mRS can assess the recovery of neurological function in patients. In this study, we used the fully automated Rapid Processing of Perfusion and Diffusion (RAPID) software (iSchemaView, Menlo Park, CA, USA) [45] to detect ischemic stroke lesions in the brain tissue. According to the detection results, the ground truth (1—ischemic stroke, 0—normal) for the ischemic stroke of each DSC-PWI image can be obtained. For NIHSS assessment, depending on the NIHSS evaluated by two experienced neurologists, we redefined a score of zero as a normal state without neurological impairment and a score greater than zero as a patient with neurological impairment (neurological impairment: NIHSS > 0, normal: NIHSS = 0). For outcome prediction, we set a poor outcome as that with a 90-day mRS greater than 2 and a good outcome as that with a 90-day mRS less than 2 (good outcome: 90-day mRS ≤ 2, poor outcome: 90-day mRS > 2).

(C)Computing DRF
The DSC-PWI datasets are 4D images composed of N 3D images with a size of S × H × W, wherein N is the total number of 3D images in each DSC-PWI image and S, H, and W represent the slice, height, and width of the 3D images, respectively. This study used radiomics technology to compute the DRF of the brain tissue in the DSC-PWI image by splitting the DSC-PWI image into N 3D images. First, by decomposing the 4D images into N (50 in this study) single 3D images, the radiomics features of the brain tissue in each 3D image could be computed separately. Then, the DRF can be obtained by combining the radiomics features of all of the 3D images at the time order in the DSC-PWI image (seen in Figure 1b). This study calculated six original feature groups and used six filters to process the original feature groups. The original feature groups were the first-order statistics (First_order), gray-level co-occurrence matrix (GLCM), gray-level run-length matrix (GLRLM), gray-level size-zone matrix (GLSZM), gray-level dependency matrix (GLDM), and neighboring gray-tone difference matrix (NGTDM). The six filters included log sigma with scale {1.0, 2.0, 3.0, 4.0, 5.0}, wavelet, square, square root, logarithm, and exponential. To categorize the features, we summarized the filtering results into the original feature group. For example, the filtered GLCM features can be summarized into the group of GLCM. Thus, the final six feature sets were obtained, including First_order, GLCM, GLRLM, GLSZM, GLDM, and NGTDM. In this study, radiomics feature calculation was automatically performed using the PyRadiomics package implemented in Python [46,47]. Each 3D image in the DSC-PWI data was defined as S(n), wherein n was from zero to 49, and the DSC-PWI image was represented as set {S(0), S(1), …, S(49)}. Moreover, the calculated DRF were renamed by connecting their original name and the n-value of the 3D image S(n). For example, “log-sigma-1-0-mm-3D_firstorder_Skewness_17” represents the radiomics feature “log-sigma-1-0-mm-3D_firstorder_Skewness” of S(17), which is the 17th 3D image in DSC-PWI data, and this feature belongs to the First_order group.

#### 2.2.2. Feature Selection and Combination Strategy

This study combined the feature selection method (least absolute shrinkage and selection operator, Lasso) and various feature dimension-reduction algorithms to explore the role of different combinations of DRF in ischemic stroke. The details are introduced in the following and shown in Figure 1c.
(A)Extracting significant DRF

Before feature selection, feature normalization is necessary to eliminate the influence of dimension and value-range differences between features. This study used each feature’s mean and standard deviation to normalize the feature vector. The transformation is given in Equation (1).
(1)$$F_{i}^{*}=(F_{i}-\bar{F_{i}})/\left(F_{imax}-F_{imin}\right)$$
where $F_{i}^{*}$ is the normalized feature of the i*th* feature $F_{i}$, and the variables $\bar{F_{i}}$, $F_{imax}$, and $F_{imin}$ are the mean, maximum, and minimum of $F_{i}$, respectively.

Then, this study used the *t*-test algorithm to extract the significant DRF from all DRF obtained in Section 2.2.1 (C). First, the homogeneity of variance test was performed to detect whether the feature has the homogeneity of variance. When the feature had homogeneity of variance, the *t*-test was performed directly. However, if the feature did not have homogeneity of variance, the parameter *equal_val = False* needed to be added during the *t*-test analysis. This study used the Levene test to realize the homogeneity of the variance test. Finally, the significant DRF with values of *p* < 0.05 in the *t*-test analysis remained to complete subsequent feature-selection processing. Therefore, according to the three evaluation items, three sets of significant DRF were obtained, and the significant DRF were defined as F*_t_*_-test_ in this study.
(B)Supervised feature selection

Feature selection aims to find the most compelling feature representing the target variable and compress the feature space. Lasso has been recognized as one of the most effective feature selection methods for selecting relevant features to the target variable [40,48,49]. This study used Lasso to select outstanding DRF depending on the three sets of ground truth, and the DRF with a non-zero coefficient were selected as the outstanding DRF. The Lasso was implemented by the *LassoCV* function imported from the *sklearn.linear_model* package in Python 3.6, and the *cv* was set as 10 in the function (seen in Table 2). The mathematical principle of Lasso is shown in Equation (2). By the supervised feature selection, three groups of outstanding DRF can be obtained according to three sets of ground truth. For each evaluation item, the selected outstanding features were defined as Lasso(F*_t_*_-test_, item) in this study.
(2)$$Lasso(F_{t-test},item)=argmin\{\sum_{i=1}^{M}{\left(y_{i}-\beta_{0}-\sum_{j=1}^{q}\beta_{j}x_{ij}\right)}^{2}+l\sum_{j=0}^{q}\left|\beta_{j}\right|\}$$
wherein Lasso (F*_t_*_-test_, item) represents the selected outstanding DRF for the evaluation item from the significant DRF F*_t_*_-test_; $x_{ij}$ is the independent DRF in F*_t_*_-test_; $y_{i}$ is the ground truth of the *i*th case; λ is the penalty parameter greater than zero; $\beta_{j}$ is the regression coefficient; *M* is the number of cases; *q* is the number of selected outstanding DRF; and i ∈ [1, M], and j ∈ [0, q].
(C)Unsupervised feature selection

This study adopted unsupervised feature selection methods to extract various additional DRF. As one of the unsupervised feature selection methods, the feature dimension-reduction algorithm (DRA) has been applied. In detail, the DRAs used in this study include principal component analysis (PCA), independent component correlation algorithm (ICA), t-distributed stochastic neighbor embedding (TSNE), uniform manifold approximation and projection (UMAP), and isometric feature mapping (ISOMAP). All of the feature dimension-reduction methods were implemented by importing the corresponding package in python 3.6 (seen in Table 2). By setting the parameter *n_components* to 10, each dimensionality-reduction method reduced the significant DRF to 10 features. The obtained dimension-reduction DRF were defined as the name of the dimension-reduction method. For example, the dimension-reduction features obtained from PCA can be defined as *PCA*, and the dimension-reduction DRF in PCA were defined as {*PCA*0, *PCA*1, …, *PCA*9}.
(D)Feature combination strategy

The feature combination strategy in this study was used to generate four experimental groups for each evaluation item (seen in Figure 2). First, experimental group_A {F*_t_*_-test_, Lasso(F*_t_*_-test_, item) and DRA (F*_t_*_-test_, item)} included significant DRF F*_t_*_-test_, outstanding DRF Lasso(F*_t_*_-test_, item), and dimension-reduction feature DRA (F*_t_*_-test_, item). Second, experimental group_B was the combination of Lasso(F*_t_*_-test_, item) and each DRA (F*_t_*_-test_, item), which can be defined as group_B {Lasso(F*_t_*_-test_, item) + DRA (F*_t_*_-test_, item)}. The signal ‘+’ means combination. Third, the experimental group_ C was the collection of the selected outstanding DRF from DRA (F*_t_*_-test_, item) by Lasso, which was defined as group_C {Lasso(DRA(F*_t_*_-test_, item), item)}. Finally, experimental group D was the combination of Lasso(F*_t_*_-test_, item) and each group_C, and the set was group D {Lasso(F*_t_*_-test_, item) + Lasso(DRA(F*_t_*_-test_, item), item)}. Based on the above, four experimental groups can be obtained for each evaluation item in this study.

#### 2.2.3. Performance Evaluation

This study used ten supervised machine learning models to fully evaluate the effectiveness of the feature sets in the four experimental groups described above (seen in Figure 1d). The area under the curve score (AUC) was applied to evaluate the classification ability of each feature set in the four experimental groups. In detail, ten-fold cross-validation was performed to compute the AUC, and the ten machine learning models include support vector machine (SVM), decision tree (DT), Adaboost classifier (Ada), neural network (NN), random forest (RF), k-nearest neighbors (KNN), logistic regression (LR), linear discriminant analysis (DA), gradient boosting classifier (GBDT), and GaussianNB (NB) (seen in Table 3).

## 3. Results

The results are divided into four sections, including preprocessing results, generated four experimental groups, and the performance of four experimental groups. The details are shown in the following.

### 3.1. Preprocessing Results

#### 3.1.1. Ground Truth Distribution for Three Evaluation Items

According to the ischemic stroke detection by RAPID, this study detected 78 (50%) ischemic stroke cases in 156 DSC-PWI images. The number of NIHSS equal to zero was 95 (60.9%), and the 90-day mRS less than 2 (good outcome) was 101 (66%) (seen in Table 4). The ground truth equal to 1 means patients with ischemic stroke lesions, neurological impairment, or poor outcome, and the ground truth equal to zero means no ischemic stroke lesions, normal neurological function, or good outcome.

#### 3.1.2. Computed DRF

For each 4D DSC-PWI image, 83700 DRF (50 3D images × 1674 features) could be calculated (seen in Figure 3a and Table 5). These DRF were divided into six groups: (1) First-order (324 features × 50 = 16,200 features), (2) GLCM (432 features × 50 = 21,600 features), (3) GLRLM (288 features × 50 = 14,400 features), (4) GLSZM (288 features × 50 = 14,400 features), (5) NGTDM (90 features × 50 = 4500 features), and (6) GLDM (252 features × 50 = 12,600 features).

### 3.2. Selected Outstanding DRF and Dimension-Reduction DRF

#### 3.2.1. Significant DRF for Three Evaluation Items

By the *t*-test analysis, this study extracted 25,822 significant DRF for ischemic stroke detection, and there were 5118 DRF in First_order, 7698 in GLCM, 3800 in GLDM, 4117 in GLRLM, 3737 in GLSZM, and 1352 in NGTDM. Their *p*-values ranged from 0.0123 ± 0.0144. Furthermore, 8324 significant DRF with p-values 0.0232 ± 0.0156 were extracted for NIHSS assessment. Among them, 2061 DRF were in First_order, 2655 in GLCM, 866 in GLDM, 1016 in GLRLM, 1289 in GLSZM, and 437 in NGTDM. Furthermore, 9203 significant DRF with *p*-values 0.0238 ± 0.0144 were extracted for outcome prediction, and there were 2089 DRF in First_order, 2650 in GLCM, 1304 in GLDM, 1254 in GLRLM, 1439 GLSZM, and 467 in NGTDM. The detailed statistics of each significant DRF were introduced in Table 5 and Figure 3b–d and Figure 4a).

#### 3.2.2. Selected Outstanding DRF for Three Evaluation Items

This study used the Lasso algorithm to select outstanding DRF from significant DRF for each evaluation item. As a result, 34 outstanding DRF were selected for ischemic stroke detection and defined as Lasso (F*_t_*_-test_, ischemic stroke detection). Among them, there were only 1 DRF in First_order, 9 in GLCM, 13 in GLDM, 5 in GLRLM, 4 in GLSZM, and 2 in NGTDM. Besides, 32 and 40 outstanding DRF were selected for NIHSS evaluation and outcome prediction, respectively. In the Lasso (F*_t_*_-test_, NIHSS evaluation), there were 7 outstanding DRF in First_order, 5 in GLCM, 5 in GLDM, 2 GLRLM, 4 in GLSZM, and 8 in NGTDM. In the Lasso (F*_t_*_-test_, Outcome prediction), there were 6 outstanding DRF in First_order, 21 in GLCM, 9 in GLDM, 2 in GLSZM, 2 in NGTDM, and none in GLRLM. All of the outstanding DRF for the three evaluation items were completely different, with only one DRF (F4) selected for both ischemic stroke detection and outcome prediction. Thus, a total of 105 outstanding DRF were selected. The outstanding DRF were renamed as F_k_, and k represents the order of outstanding DRF.

Besides, the absolute values of Pearson correlation coefficients (R-values) between outstanding DRF and ground truths for ischemic stroke detection ranged from 0.158 to 0.451. For NIHSS evaluation, the absolute R-values between outstanding DRF and ground truths for NIHSS evaluation ranged from 0.156 to 0.365, and the absolute R-values between outstanding DRF and ground truths for outcome prediction ranged from 0.161 to 0.334. The results showed that the selected outstanding DRF had weak or moderate correlations with ischemic stroke, NIHSS, and outcome (seen in Figure 4b–d).

#### 3.2.3. Dimension-Reduction DRF Obtained from Five Dimension-Reduction Algorithms

This study’s feature dimension-reduction algorithms are unsupervised feature selection methods, and each feature dimension-reduction algorithm can obtain the same set of dimensionality-reduction DRF for the three evaluation items. The R-value of each dimension-reduction DRF with the corresponding ground truth was calculated and is shown in Figure 5. For example, for ischemic stroke detection, the dimension-reduction DRF obtained from PCA had R-values of 0.110 ± 0.121 and that from ICA, TSNE, IOSMAP, and UMAP had R-values of 0.140 ± 0.079, 0.110 ± 0.121, 0.294 ± 0.139, and 0.098 ± 0.133, respectively. For NIHSS evaluation, the dimension-reduction DRF obtained from PCA, ICA, TSNE, IOSMAP, and UMAP had R-values of 0.097 ± 0.089, 0.107 ± 0.075, 0.088 ± 0.069, 0.077 ± 0.052, and 0.155 ± 0.113. For outcome prediction, the dimension-reduction features obtained from PCA, ICA, TSNE, IOSMAP, and UMAP had R-values of 0.093 ± 0.092, 0.097 ± 0.087, 0.093 ± 0.092, 0.100 ± 0.088, and 0.157 ± 0.113.

#### 3.2.4. Selected Outstanding Dimension-Reduction DRF

This study used the Lasso algorithm to extract the outstanding dimension-reduction DRF from the dimension-reduction DRF in PCA, ICA, TSNE, ISMOP, and UMAP. As a result (seen in Figure 6), none of the dimension-reduction DRF in ICA were selected for the three evaluation items. Besides, for PCA, two outstanding dimension-reduction DRF (PCA0 and PCA1) for NIHSS evaluation and outcome prediction and four outstanding dimension-reduction DRF (PCA0, PCA1, PCA3, PCA4) for ischemic stroke detection were selected. Similar results were achieved for TSNE. TSNE0 and TSNE1 were selected for NIHSS evaluation and outcome prediction, and additional TSNE3 and TSNE4 were selected for ischemic stroke detection. For UMAP and ISOMAP, no outstanding dimension-reduction DRF was selected for outcome prediction. Furthermore, {UMAP0, UMAP3, UMAP4} and {ISOMAP0, ISOMAP1, ISOMAP5} were selected for ischemic stroke detection. UMAP1 and {ISOMAP0, ISOMAP1, ISOMAP7} were selected for NIHSS evaluation.

### 3.3. Performance of Four Experimental Groups

This study evaluated the performance of the four constructed experimental groups in classifying and predicting neurological impairment by ischemic stroke detection, NIHSS evaluation, and outcome prediction. Among the four experiments, the combination DRF in experimental group_D achieved the best score in the three evaluation items. In particular, the combinations of Lasso + PCA_Lasso and Lasso + TSNE_Lasso achieved the same highest score, respectively, which proved the potentially significant value of DRF in the clinical treatment and prognosis analysis of stroke. The detailed results were introduced as follows.

For experimental group_A (seen in Table 6), the outstanding DRF from Lasso performed best, and the five dimension-reduction DRF achieved a similar score, with the significant DRF F*_t_*_-test_ selected by *t*-test analysis in the three items. In detail, the significant DRF achieved the best AUC of 0.731 for ischemic stroke detection, 0.652 for NIHSS assessment, and 0.679 for outcome prediction. In terms of ischemic stroke detection, the AUCs of dimension-reduction DRF obtained by PCA, TSNE, UMAP, ICA, and ISOMAP ranged from 0.672 ± 0.033, 0.664 ± 0.036, 0.681 ± 0.040, 0.684 ± 0.034, and 0.687 ± 0.027, respectively. In contrast, all of the AUCs of outstanding DRF selected by Lasso were better than 0.731, ranging from 0.819 ± 0.064, and the best AUC was 0.837. In terms of NIHSS assessment, the AUCs of dimension-reduction DRF obtained by PCA, TSNE, UMAP, ICA, and ISOMAP ranged from 0.540 ± 0.038, 0.529 ± 0.025, 0.502 ± 0.050, 0.531 ± 0.045, and 0.551 ± 0.050, and the best one of them had an AUC of 0.649. The outstanding DRF from Lasso achieved the best AUC of 0.795, and the performance of significant DRF (best AUC = 0.652) was better than all of the dimension-reduction DRF (best AUC = 0.649). In terms of outcome prediction, the AUCs of dimension-reduction DRF obtained by PCA, TSNE, UMAP, ICA, and ISOMAP ranged from 0.544 ± 0.025, 0.544 ± 0.025, 0.560 ± 0.049, 0.562 ± 0.029, and 0.602 ± 0.036. Similarly, the best AUC of 0.818 was achieved by outstanding DRF from Lasso, and the performance of significant DRF (best AUC = 0.679) was better than all of the dimension-reduction DRF (best AUC = 0.646).

For experimental group_B (Table 7), the performance can be improved when combining the outstanding DRF with each dimension-reduction DRF in ischemic stroke detection and NIHSS assessment. However, these combinations failed in outcome prediction. In detail, in terms of ischemic stroke detection, when combining Lasso with PCA, the best AUC increased from 0.711 (PCA) to 0.899 (Lasso + PCA); when combining Lasso with TNSE, the best AUC increased from 0.710 (TNSE) to 0.905 (Lasso + TNSE); when combining Lasso with UMAP, the best AUC increased from 0.723 (UMAP) to 0.873 (Lasso + UMAP); when combining Lasso with ICA, the best AUC increased from 0.739 (ICA) to 0.893 (Lasso + ICA); when combining Lasso with ISOMAP, the AUC increased from 0.731 (ISOMAP) to 0.874 (Lasso + ISOMAP). Besides, the above combination performed better than the best score of 0.873 (Lasso) in experimental group_A. In terms of NIHSS assessment, when combining Lasso with PCA and TSNE, the best AUC increased from 0.583 (PCA, TSNE) to 0.835 (Lasso + PCA, Lasso + TSNE); when combining Lasso with UMAP, the best AUC increased from 0.618 (UMAP) to 0.786 (Lasso + UMAP); when combining Lasso with ICA, the best AUC increased from 0.610 (ICA) to 0.835 (Lasso + ICA); when combining Lasso with ISOMAP, the AUC increased from 0.668 (ISOMAP) to 0.812 (Lasso + UMAP). Besides, the Lasso + UMAP obtained a lower score than the best score of 0.795 (Lasso) in experimental group_A, and the other four combinations achieved better scores. Regarding outcome prediction, when combining Lasso with PCA, TSNE, and ICA, the best AUCs were 0.806; when combining Lasso with UMAP and ISOMAP, the best AUCs were 0.795 and 0.814, respectively. In this item, all of the combinations of Lasso and each dimension-reduction DRF were inferior to the outstanding DRF (Lasso) but better than the single dimension-reduction DRF and significant DRF (*t*-test) in experimental group_A.

For experimental group_C (Table 8), the selected outstanding dimension-reduction DRF (PCA_Lasso, TSNE_Lasso, UMAP_Lasso, and IOSMAP_Lasso) achieved a better AUC than the dimension-reduction DRF in experimental group_A. However, it was still inferior to the Lasso in experimental group_A and the combinations in experimental group_B. In detail, in terms of ischemic stroke detection, the best AUCs of the outstanding dimension-reduction DRF in PCA, TSNE, UMAP, and ISOMAP were 0.742, 0.742, 0.748, and 0.742, which were better than the AUCs of the original dimension-reduction DRF in experimental group_A, 0.711, 0.710, 0.723, and 0.731, but lower than the best performance, 0.873, in experimental group_A and 0.905 in experimental group_B. In terms of NIHSS assessment, the best AUCs of the outstanding dimension-reduction DRF in PCA, TSNE, and UMAP (0.645, 0.660, and 0.536) were better than the AUCs of the original dimension-reduction DRF (0.619, 0.568, and 0.610) in experimental group_A. In contrast, the best AUC (0.581) of the outstanding dimension-reduction DRF in ISOMAP was inferior to the AUC (0.649) of ISOMAP in experimental group_A. The best AUC in experimental group_C was lower than in experimental group_A and experimental group_B. Regarding outcome prediction, the best AUC’s 0.596 of outstanding dimension-reduction DRF in PCA and TSNE was better than that of the original dimension-reduction DRF in experimental group_A but inferior to the best performance 0.818 in experimental group_A and 0.814 in experimental group_B.

For experimental group_D (Table 9), the combination of outstanding DRF and outstanding dimension-reduction DRF provided a chance to improve their performance further. In terms of ischemic stroke detection, the combination of outstanding DRF and outstanding dimension-reduction DRF in PCA, TSNE, UMAP, and ISOMAP obtained the best AUCs, 0.925, 0.925, 0.873, and 0.887, which were the highest scores among all of the experimental groups. In terms of NIHSS assessment, the combination of outstanding DRF and outstanding dimension-reduction DRF in PCA and TSNE achieved the best AUC of 0.853 in all four experimental groups. The combination of outstanding DRF and outstanding dimension-reduction DRF in UMAP and ISOMAP failed to surpass the best performance in experimental group_B, with the best AUC of 0.787 and 0.829, respectively. Finally, regarding outcome prediction, the combination of Lasso and outstanding dimension-reduction DRF in PCA and TSNE achieved the best AUC of 0.828, which also was the best score in the four experimental groups. Besides, among all of the ten classification models, the performance of DA and LR was better than the other models, and they achieved almost all of the best AUCs.

## 4. Discussion

The ability to accurately detect ischemic stroke and predict its neurological recovery is of great clinical value [50], which can help to prepare appropriate treatment plans and improve the prognosis and recovery of patients. Previous studies have proved the value of medical images (CT, DWI, SWI, PWI, CTP, and ASL) in ischemic stroke detection [51,52,53,54,55], outcome prediction [56,57,58], and the association between NIHSS and prognosis [59,60,61]. In addition, some scholars have introduced radiomics technology into the above studies [36,40,41]. However, these studies generally focused on the relationship between local features in lesions while ignoring global information in the whole brain. Furthermore, few studies directly used the dynamic perfusion information in DSC-PWI images to perform the above works. This study intended to explore the role of whole-brain DRF in evaluating the state of brain tissue and neurological function, especially in ischemic stroke diagnosis, the assessment of neurological impairment, and outcome prediction. As a result, the highest AUC was 0.925 for ischemic stroke diagnosis, 0.846 for the assessment of neurological impairment (NIHSS), and 0.835 for outcome prediction (90-day mRS). Therefore, the reproducibility and applicability of this study indicate the feasibility of whole-brain DRF-based radiomics in detecting and assessing ischemic stroke and predicting the neurological recovery of ischemic stroke patients.

This paper is an exploratory work based on the DRF of the whole brain. In DSC-PWI images, the intensity of voxels changes under the action of the contrast agent, resulting in changes in 3D image features. Therefore, the 3D image features can reflect the process of the propagation of the contrast agent in the brain tissue and then indicate the state of blood flow propagation. Furthermore, ischemic stroke is a vascular disease that causes tissue and neurological damage due to vascular blockage. Therefore, there is an inevitable relationship between the extracted time-related DRF and ischemic stroke. Some studies reported that the intensity drop of the ischemic tissue was less than normal tissue, and the decline rate was slower than in normal tissues [62]. In addition, although ischemic stroke is caused by local tissue ischemia, when the local tissue is in an ischemic state, the blood flow characteristics of the whole brain will also change accordingly, since blood flow propagation is carried out in the whole brain. Therefore, global cerebral blood flow features may provide a chance to diagnose and treat ischemic stroke. Based on the above, this study used radiomics technology to extract the radiomics features of each 3D image in the time series of DSC-PWI images. Then DRF that reflects the blood flow transmission changes of the whole brain were generated. From the results of this study, the original significant DRF F*_t_*_-test_ can obtain an AUC of 0.731 for ischemic stroke, 0.652 for NIHSS assessment, and 0.679 for outcome prediction. It can be seen that DRF can reflect the changes in blood flow and assess the state of brain tissue and the degree of damage to brain nerve function.

Lasso and dimension-reduction algorithms have been used in various scenes of ischemic stroke analysis [48,49,63,64]. This study used Lasso and five dimension-reduction algorithms to generate four experimental groups, evaluating the role of DRF in the diagnosis, assessment, and outcome prediction of ischemic stroke. Based on the results in experimental group_A, the performance of outstanding DRF from Lasso was better than dimension-reduction DRF from PCA, ICA, TSNE, UMAP, and ISOMAP, and the best score of dimension-reduction DRF was similar to the original significant DRF from T-test. Therefore, it may mean that Lasso was the best among all of the feature selection methods. Then, this study combined the outstanding DRF from Lasso with each dimension-reduction DRF from PCA, ICA, TSNE, UMAP, and ISOMAP and evaluated them in experimental group_B. Based on the results in experimental group_B, the combination features achieved a better AUC than single features in experimental group_A for stroke diagnosis and NIHSS assessment and failed to surpass single features in experimental group_A for outcome prediction. Besides, this study used Lasso to select outstanding DRF from dimension-reduction DRF (PCA, ICA, TSNE, UMAP, and ISOMA), which were shown as experimental group_C. Based on the results in experimental group_C, the performance of outstanding DRF in experimental group_A performed better than outstanding dimension-reduction DRF in PCA, ICA, TSNE, UMAP, and ISOMAP, and the best score of dimension-reduction DRF was similar to the original significant DRF from *t*-test. This also means that Lasso was the best one among all of the feature selection methods. Finally, the outstanding DRF in experimental group_A and outstanding dimension-reduction DRF in experimental group_C were combined, achieving the best score for the three evaluation items. Thus, it can be concluded that different combinations of Lasso and dimension-reduction algorithms could achieve different results for stroke detection, NIHSS assessment, and outcome prediction.

Although the results showed that the combination of outstanding DRF and outstanding dimension-reduction DRF selected from Lasso could improve performance in stroke detection, NIHSS assessment, and outcome prediction, further optimization of the models can be regarded as one future work. We will validate our improved method with the more extensive and varied datasets before applying it to clinical trials in future work. The results in this study do not mean that the models can be used alone for stroke treatment decision-making. Instead, this study should be considered a support tool in stroke treatment guidance.

## 5. Conclusions

In conclusion, although different feature selection methods on whole-brain DRF achieved diverse performance, the critical role of whole-brain DRF in ischemic stroke detection, NIHSS assessment, and outcome prediction has been proven. From experimental group_A to experimental group_D, it can be concluded that the combination of outstanding DRF with outstanding dimension-reduction DRF in experimental group_D performed best in all of the three evaluation items. Comparing the best AUC of F*_t_*_-test_ in experimental group_A and the best_AUC experimental group_D, the AUC in stroke detection increased by 19.4% (from 0.731 to 0.925), the AUC in NIHSS assessment increased by 20.1% (from 0.652 to 0.853), and the AUC in prognosis prediction increased by 14.9% (from 0.679 to 0.828). This study provided a potential clinical tool for detailed clinical diagnosis and outcome prediction before treatment.

## Figures and Tables

**Figure 1 jcm-11-05364-f001:**
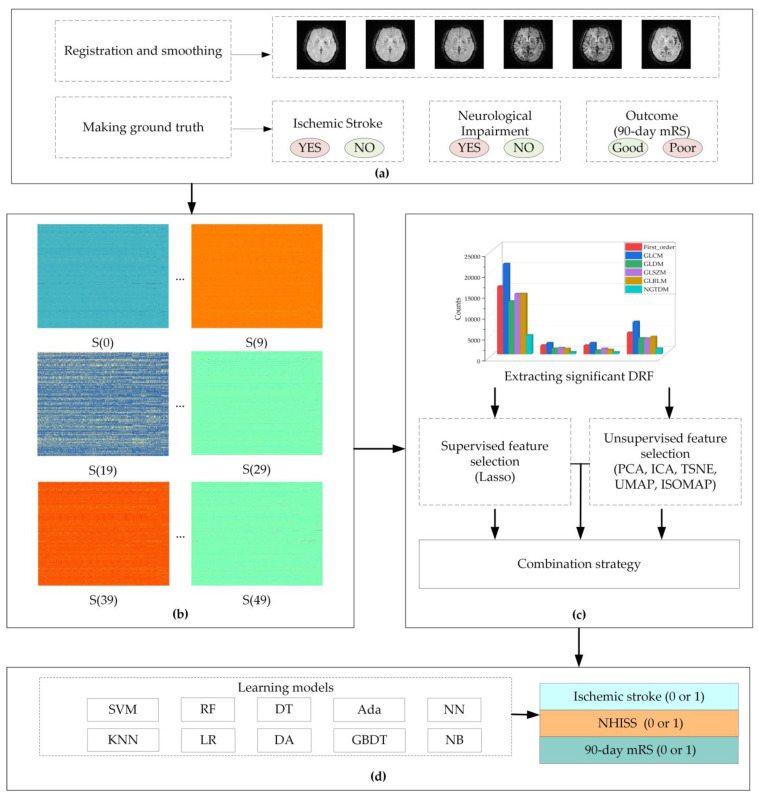
Flowchart of this study. (**a**) preprocessing of dynamic susceptibility contrast perfusion-weighted imaging (DSC-PWI) datasets; (**b**) computing whole-brain dynamic radiomics features (DRF); (**c**) feature selection and combination strategy; (**d**) evaluating performance with ten learning models. The DRF in (**b**) are combined with the radiomics features of 3D images in the time series of DSC-PWI image; the five unsupervised feature selection are principal component analysis (PCA), independent component correlation algorithm (ICA), t-distributed stochastic neighbor embedding (TSNE), uniform manifold approximation and projection (UMAP), and isometric feature mapping (ISOMAP); the ten models are support vector machine (SVM), decision tree (DT), Adaboost classifier (Ada), neural network (NN), random forest (RF), k-nearest neighbors (KNN), logistic regression (LR), linear discriminant analysis (DA), gradient boosting classifier (GBDT), and GaussianNB (NB).

**Figure 2 jcm-11-05364-f002:**
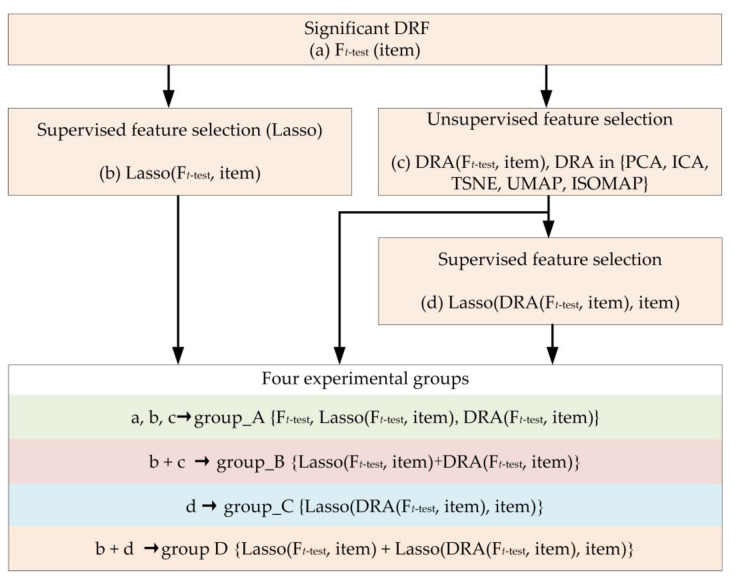
The flowchart of the feature combination strategy in our study.

**Figure 3 jcm-11-05364-f003:**
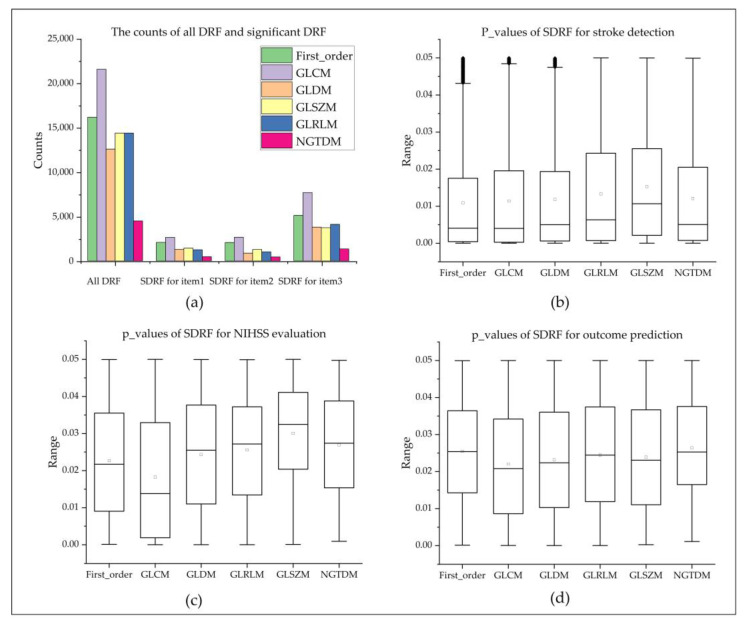
Statistics of all DRF and box plots of DRF for three evaluation items. (**a**) shows the distribution of DRF; (**b**–**d**) are the box plots of the p-values of significant DRF in each feature group for stroke detection, NIHSS evaluation, and outcome prediction, wherein item 1, item 2, and item 3 are stroke detection, NIHSS evaluation, and outcome prediction, respectively.

**Figure 4 jcm-11-05364-f004:**
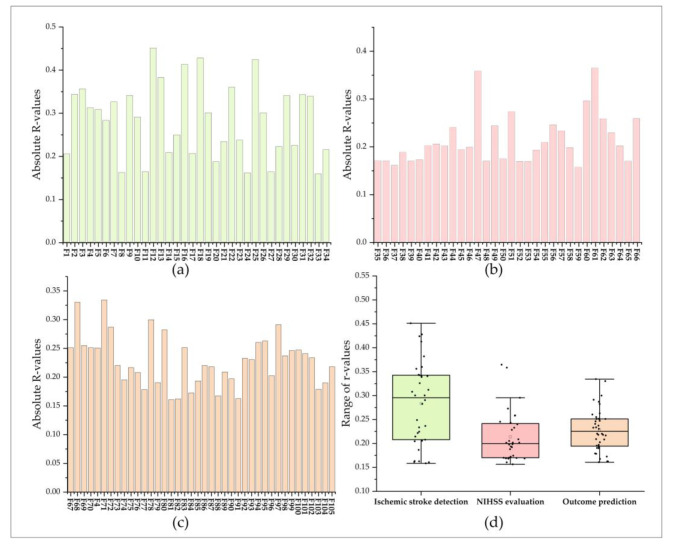
Correlation between outstanding DRF and the ground truths of three evaluation items. (**a**–**c**) are the Pearson correlation coefficients between outstanding DRF with ground truth for ischemic stroke detection, NIHSS evaluation, and outcome prediction; (**d**) is a box plot of the Pearson correlation coefficients for the three evaluation items.

**Figure 5 jcm-11-05364-f005:**
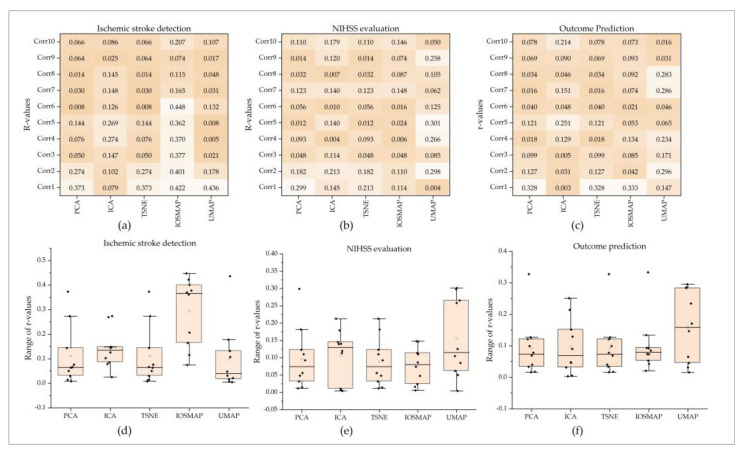
Dimension-reduction DRF for the three evaluation items. (**a**–**c**) are the Pearson correlation coefficients between dimension-reduction DRF and the ground truth for ischemic stroke detection, NIHSS evaluation, and outcome prediction. (**d**–**f**) are box plots of the Pearson correlation coefficients for the three evaluation items.

**Figure 6 jcm-11-05364-f006:**
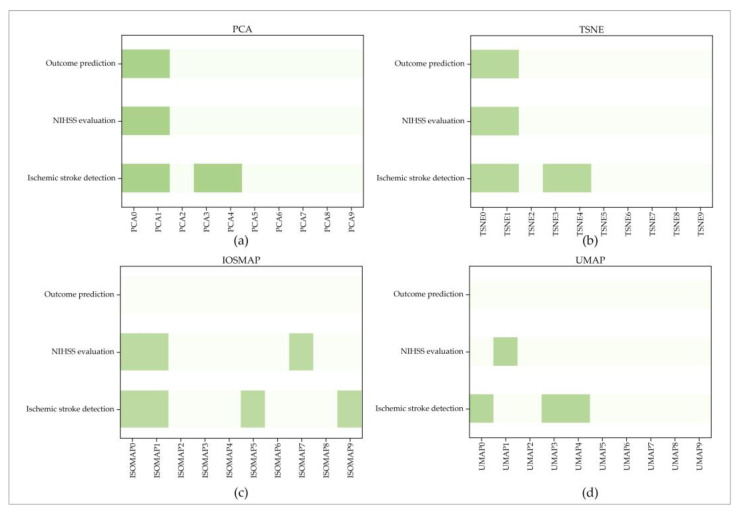
Outstanding dimension-reduction DRF for the three evaluation items. (**a**–**d**) are the selected outstanding dimension-reduction DRF by PCA, TSNE, ISOMAP, and UMAP for the three evaluation items; the dark green represents the selected outstanding dimension-reduction DRF.

**Table 1 jcm-11-05364-t001:** Patient information and scanning parameters of DSC-PWI datasets.

Patient Information		Scanning Parameters of DSC-PWI Images	
Numbers of patients	88	TE/TR	32/1590 ms
Datasets (sets)	156	Matrix	256 × 256
Image with ischemic stroke (%)	78 (50%)	FOV	230 × 230 mm^2^
image of outcome patient (%)	73 (46.8%)	Thickness	5 mm
Female (%)	39 (25%)	Number of measurements	50
Age (Mean ± Std)	9.919 ± 6.747	Spacing between slices	6.5 mm
NIHSS (Mean ± Std)	6.275 ± 6.875	Pixel bandwidth	1347 Hz/pixel
90-day mRS	38 (47.5%)	Number of slices	20

**Table 2 jcm-11-05364-t002:** The implementation of feature selection and feature dimension reduction.

Method	Implementation in Python 3.6
Lasso	LassoCV (alphas = alphas, cv = 10, max_iter = 100,000, normalize = False). fit (features, targets)
PCA	sklearn.decomposition.PCA (svd_solver = ‘auto’, n_components = num_fea)
ICA	sklearn.decomposition.FastICA (n_components = num_fea, random_state = 12, max_iter = 1,000,000)
tSNE	sklearn.manifold.TSNE (n_components = num_fea, init = ‘pca’, random_state = 12, method = ‘exact’)
UMAP	umap.UMAP (n_neighbors = 5, min_dist = 0.3, n_components = num_fea). fit_transform(features)
ISOMAP	sklearn.manifold.Isomap (n_neighbors = 5, n_components = num_fea, n_jobs = −1). fit_transform(features)

**Table 3 jcm-11-05364-t003:** Descriptions of the 10 models in this study.

Model	Definition in Python 3.6
SVM	sklearn.svm.SVC (kernel = ‘rbf’, probability = True)
DT	sklearn.tree. DecisionTreeClassifier ()
Ada	sklearn.ensemble.AdaBoostClassifier ()
NN	sklearn.neural_network. MLPClassifier (hidden_layer_sizes = (400, 100), alpha = 0.01, max_iter = 10,000)
RF	sklearn.ensemble.RandomForestClassifier (n_estimators = 200)
KNN	sklearn.neighbors. sklearn.neighbors ()
LR	sklearn.linear_model.logisticRegressionCV(max_iter = 100,000, solver = “liblinear”)
DA	sklearn.discriminant_analysis ()
GBDT	sklearn.ensemble.GradientBoostingClassifier ()
NB	sklearn.naive_bayes. GaussianNB ()

**Table 4 jcm-11-05364-t004:** Ground truth distribution for three evaluation items.

Ground Truth	Ischemic Stroke	NIHSS	90-Day mRS
1	78	61	55
0	78	95	101

**Table 5 jcm-11-05364-t005:** Significant DRF statistics for the three evaluated items.

Item	Feature Group	Significant DRF	Mean	Std	Min	Medium	Max
Stroke detection	First_order	5118	0.0109	0.0139	<0.0001	0.0041	0.0500
	GLCM	7698	0.0114	0.0143	<0.0001	0.0040	0.0500
	GLDM	3800	0.0118	0.0142	<0.0001	0.0050	0.0500
	GLRLM	4117	0.0133	0.0149	<0.0001	0.0063	0.0500
	GLSZM	3737	0.0153	0.0148	<0.0001	0.0107	0.0500
	NGTDM	1352	0.0121	0.0145	<0.0001	0.0050	0.0499
NIHSS evaluation	First_order	2061	0.0227	0.0146	0.0001	0.0217	0.0500
	GLCM	2655	0.0183	0.0166	<0.0001	0.0138	0.0500
	GLDM	866	0.0243	0.0154	<0.0001	0.0255	0.0500
	GLRLM	1016	0.0256	0.0144	<0.0001	0.0272	0.0499
	GLSZM	1289	0.0300	0.0135	0.0001	0.0325	0.0500
	NGTDM	437	0.0269	0.0136	0.0009	0.0274	0.0498
Outcome prediction	First_order	2089	0.0255	0.0137	0.0001	0.0254	0.0499
	GLCM	2650	0.0220	0.0147	<0.0001	0.0208	0.0500
	GLDM	1304	0.0232	0.0148	<0.0001	0.0224	0.0500
	GLRLM	1254	0.0244	0.0147	<0.0001	0.0244	0.0500
	GLSZM	1439	0.0239	0.0143	0.0002	0.0231	0.0500
	NGTDM	467	0.0264	0.0127	0.0011	0.0253	0.0500

**Table 6 jcm-11-05364-t006:** The performance of DRF in experimental group_A.

Item	Classifier	Lasso	PCA	TSNE	UMAP	ICA	IOSMAP	*t*-Test
Stroke detection	SVM	0.861	0.691	0.691	0.710	0.686	0.731	0.716
	nn	0.861	0.680	0.634	0.680	0.666	0.646	0.713
	RF	0.783	0.711	0.666	0.660	0.688	0.704	0.698
	DT	0.662	0.615	0.596	0.613	0.632	0.647	0.601
	KNN	0.873	0.669	0.669	0.692	0.666	0.705	0.719
	Ada	0.797	0.642	0.642	0.639	0.739	0.669	0.679
	LR	0.854	0.704	0.704	0.723	0.729	0.699	0.723
	NB	0.840	0.639	0.639	0.722	0.646	0.685	0.653
	GBDT	0.791	0.659	0.685	0.648	0.677	0.683	0.687
	DA	0.867	0.710	0.710	0.722	0.710	0.692	0.731
NIHSS evaluation	SVM	0.727	0.482	0.482	0.500	0.528	0.492	0.500
	nn	0.743	0.619	0.562	0.464	0.610	0.533	0.652
	RF	0.692	0.548	0.521	0.486	0.504	0.533	0.587
	DT	0.663	0.587	0.568	0.521	0.494	0.557	0.603
	KNN	0.677	0.523	0.523	0.428	0.522	0.480	0.536
	Ada	0.731	0.526	0.526	0.491	0.476	0.597	0.574
	LR	0.795	0.532	0.532	0.496	0.512	0.568	0.629
	NB	0.755	0.527	0.527	0.618	0.607	0.649	0.667
	GBDT	0.687	0.513	0.509	0.531	0.518	0.527	0.606
	DA	0.783	0.541	0.541	0.489	0.541	0.574	0.607
Outcome prediction	SVM	0.818	0.546	0.546	0.500	0.549	0.573	0.576
	nn	0.766	0.551	0.554	0.635	0.605	0.646	0.647
	RF	0.684	0.553	0.526	0.572	0.540	0.588	0.578
	DT	0.595	0.515	0.525	0.596	0.550	0.592	0.521
	KNN	0.694	0.592	0.592	0.584	0.553	0.667	0.638
	Ada	0.681	0.504	0.504	0.525	0.562	0.622	0.553
	LR	0.797	0.546	0.546	0.503	0.510	0.559	0.679
	NB	0.818	0.526	0.526	0.616	0.606	0.597	0.664
	GBDT	0.681	0.544	0.561	0.562	0.580	0.618	0.593
	DA	0.676	0.563	0.563	0.508	0.563	0.556	0.571

**Table 7 jcm-11-05364-t007:** The performance of DRF in experimental group_B.

	Classifier	Lasso + PCA	Lasso + TSNE	Lasso + UMAP	Lasso + ICA	Lasso + IOSMAP
Stroke detection	SVM	0.691	0.691	0.840	0.861	0.731
	nn	0.730	0.685	0.848	0.842	0.743
	RF	0.808	0.802	0.796	0.808	0.809
	DT	0.684	0.700	0.648	0.682	0.641
	KNN	0.669	0.669	0.853	0.873	0.705
	Ada	0.766	0.766	0.772	0.784	0.753
	LR	0.899	0.905	0.873	0.874	0.874
	NB	0.847	0.847	0.820	0.839	0.846
	GBDT	0.790	0.790	0.778	0.758	0.777
	DA	0.893	0.893	0.843	0.893	0.837
NIHSS evaluation	SVM	0.482	0.482	0.658	0.727	0.492
	nn	0.586	0.636	0.782	0.780	0.573
	RF	0.674	0.684	0.665	0.662	0.667
	DT	0.633	0.650	0.607	0.657	0.660
	KNN	0.536	0.536	0.622	0.677	0.480
	Ada	0.701	0.701	0.676	0.684	0.747
	LR	0.824	0.824	0.776	0.805	0.800
	NB	0.760	0.760	0.732	0.745	0.744
	GBDT	0.667	0.667	0.684	0.686	0.671
	DA	0.835	0.835	0.786	0.835	0.812
Outcome prediction	SVM	0.555	0.555	0.732	0.818	0.573
	nn	0.616	0.616	0.793	0.770	0.697
	RF	0.662	0.684	0.694	0.657	0.669
	DT	0.634	0.618	0.623	0.673	0.622
	KNN	0.594	0.594	0.639	0.694	0.667
	Ada	0.715	0.715	0.696	0.689	0.697
	LR	0.770	0.770	0.795	0.802	0.765
	NB	0.804	0.804	0.795	0.796	0.814
	GBDT	0.642	0.654	0.697	0.663	0.660
	DA	0.806	0.806	0.735	0.806	0.716

**Table 8 jcm-11-05364-t008:** The performance of DRF in experimental group_C.

	Classifier	PCA_Lasso	TSNE_Lasso	UMAP_Lasso	IOSMAP_Lasso
Stroke detection	SVM	0.717	0.717	0.722	0.724
	nn	0.634	0.633	0.704	0.707
	RF	0.670	0.677	0.705	0.742
	DT	0.618	0.598	0.652	0.672
	KNN	0.704	0.704	0.690	0.679
	Ada	0.647	0.647	0.588	0.704
	LR	0.742	0.742	0.748	0.712
	NB	0.658	0.658	0.737	0.717
	GBDT	0.665	0.633	0.665	0.716
	DA	0.730	0.730	0.730	0.718
NIHSS evaluation	SVM	0.496	0.496	0.500	0.501
	nn	0.550	0.613	0.524	0.557
	RF	0.645	0.655	0.536	0.502
	DT	0.602	0.627	0.536	0.534
	KNN	0.621	0.621	0.472	0.581
	Ada	0.618	0.618	0.433	0.532
	LR	0.550	0.550	0.482	0.574
	NB	0.570	0.570	0.492	0.580
	GBDT	0.641	0.660	0.507	0.514
	DA	0.541	0.541	0.508	0.554
Outcome prediction	SVM	0.532	0.532		
	nn	0.573	0.575		
	RF	0.571	0.556		
	DT	0.524	0.506		
	KNN	0.555	0.555		
	Ada	0.505	0.505		
	LR	0.596	0.596		
	NB	0.577	0.577		
	GBDT	0.531	0.531		
	DA	0.591	0.591		

**Table 9 jcm-11-05364-t009:** The performance of DRF in experimental group_D.

	Classifier	Lasso + PCA_Lasso	Lasso + Tsne_Lasso	Lasso + UMAP_Lasso	Lasso + Iosmap_Lasso
Stroke detection	SVM	0.717	0.717	0.872	0.724
	nn	0.795	0.803	0.861	0.732
	RF	0.809	0.802	0.796	0.814
	DT	0.623	0.663	0.614	0.662
	KNN	0.704	0.704	0.834	0.679
	Ada	0.776	0.776	0.773	0.766
	LR	0.905	0.905	0.873	0.887
	NB	0.847	0.847	0.833	0.847
	GBDT	0.790	0.778	0.772	0.770
	DA	0.925	0.925	0.862	0.874
NIHSS evaluation	SVM	0.496	0.496	0.761	0.501
	nn	0.788	0.735	0.777	0.741
	RF	0.703	0.692	0.656	0.684
	DT	0.675	0.612	0.636	0.630
	KNN	0.616	0.616	0.662	0.577
	Ada	0.726	0.726	0.756	0.686
	LR	0.846	0.846	0.770	0.829
	NB	0.759	0.759	0.736	0.740
	GBDT	0.673	0.673	0.672	0.657
	DA	0.853	0.853	0.787	0.822
Outcome prediction	SVM	0.522	0.522		
	nn	0.808	0.808		
	RF	0.669	0.699		
	DT	0.592	0.615		
	KNN	0.541	0.541		
	Ada	0.705	0.705		
	LR	0.803	0.803		
	NB	0.828	0.828		
	GBDT	0.629	0.647		
	DA	0.756	0.756		

## Data Availability

The data supporting this study’s findings are available from the corresponding author upon reasonable request.

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
