# Peer review of "A Focus on the Role of DSC-PWI Dynamic Radiomics Features in Diagnosis and Outcome Prediction of Ischemic Stroke"

_jcm, 2022, doi:10.3390/jcm11185364_

Round 1

Reviewer 1 Report

Manuscript deals with ischemic stroke detection and its neurological recovery prediction by means of using different combination of selection methods with four experimental groups generation. The results of the study showed feasibility of whole-brain DRF-based radiomics in detecting and assessing ischemic stroke and predicting recovery, which is, obviously of great clinical value. Manuscript describes state-of-the-art methods and most of the cited references were published less than 5 years ago. The methods are adequately presented although the paper would probably benefit from the more detailed description of the feature groups and filters. The results are presented clearly. The conclusion that whole-brain DRF performance could be an additional tool for detailed diagnosis and outcome prediction is supported by the results.

However there are several minor issues to be corrected.

p.5. line 197: poor outcome is supposed to be les than 2

p.6 line 216: "in the original feature group" looks like accidental doubling

Author Response

Dear Editors and Reviewers:
We gratefully thank the editor and all reviewers for their time spent making their constructive
remarks and valuable suggestions. We have revised our manuscript after carefully reading your
professional comments. The responses to the reviewer's comments are as follows:
1. p.5. line 197: the poor outcome is supposed to be less than 2
Response: We apologize for the mistake. We have corrected the sentence to "we set poor outcome
with 90-day mRS greater than 2 and good outcome with 90-day mRS less than 2 (good outcome:
90-day mRS ≤2, poor outcome: 90-day mRS >2 )".
2. p.6 line 216: "in the original feature group" looks like accidental doubling
Response: We have deleted the duplicate words and clearly illustrated how features are grouped.
Thank you very much for affirming and recognizing the proposed methods in our manuscript.

Reviewer 2 Report

Thank you for this paper, I do not understand the main question of this work, I think it is too much information, perhaps you can focus on one.

line 62 does PWI include CT-Perfusion? 

line 158 how many stroke patients at all 2013-2016?

Discussion: I do not understand the main conclusion. What is the benefit of your results? I thinks you should demonstrate your results and conclusions more simple and focused

Round 2

Reviewer 2 Report

Dear authors, thank you for your responses about the study "A Focus on the Role of DSC-PWI Dynamic Radiomics Features in Diagnosis and Outcome Prediction of Ischemic Stroke". In my opinion the paper and conclusion is now more focused and my questions are sufficient. Thank you for this interesting paper.